# Lipophilic Studies and In Silico ADME Profiling of Biologically Active 2-Aminothiazol-4(5*H*)-one Derivatives

**DOI:** 10.3390/ijms241512230

**Published:** 2023-07-31

**Authors:** Małgorzata Redka, Szymon Baumgart, Daria Kupczyk, Tomasz Kosmalski, Renata Studzińska

**Affiliations:** 1Department of Organic Chemistry, Faculty of Pharmacy, Collegium Medicum in Bydgoszcz, Nicolaus Copernicus University in Toruń, 2 Jurasza Str., 85-089 Bydgoszcz, Poland; mredka@cm.umk.pl (M.R.); sz.baumgart@cm.umk.pl (S.B.); tkosm@cm.umk.pl (T.K.); 2Department of Medical Biology and Biochemistry, Faculty of Medicine, Collegium Medicum in Bydgoszcz, Nicolaus Copernicus University in Toruń, 24 Karłowicza Str., 85-092 Bydgoszcz, Poland; dariak@cm.umk.pl

**Keywords:** pseudothiohydantoin derivatives, lipophilicity, ADME analysis, liquid chromatography, TLC chromatography

## Abstract

Pseudothiohydantoin derivatives have a wide range of biological activities and are widely used in the development of new pharmaceuticals. Lipophilicity is a basic, but very important parameter in the design of potential drugs, as it determines solubility in lipids, nonpolar solvents, and makes it possible to predict the ADME profile. The aim of this study was to evaluate the lipophilicity of 28 pseudothiohydantoin derivatives showing the inhibition of 11β-hydroxysteroid dehydrogenase type 1 (11β-HSD1) using chromatographic methods. Experimentally, lipophilicity was determined by reverse phase thin layer chromatography (RP-TLC) and reverse phase high-performance liquid chromatography (RP-HPLC). In both methods, methanol was used as the organic modifier of the mobile phase. For each 2-aminothiazol-4(5*H*)-one derivative, a relationship was observed between the structure of the compound and the values of the lipophilicity parameters (log k_w_, R_M0_). Experimental lipophilicity values were compared with computer calculated partition coefficient (logP) values. A total of 27 of the 28 tested compounds had a lipophilicity value < 5, which therefore met the condition of Lipinski’s rule. In addition, the in silico ADME assay showed favorable absorption, distribution, metabolism, and excretion parameters for most of the pseudothiohydantoin derivatives tested. The study of lipophilicity and the ADME analysis indicate that the tested compounds are good potential drug candidates.

## 1. Introduction

Pseudothiohydantoin (2-aminothiazol-4(5*H*)-one) is a partially hydrogenated thiazole derivative containing an amino group in the 2-position and a carbonyl group in the 4-position. Pseudothiohydantoin derivatives exhibit a number of biological activities. Over the years, these compounds have been studied in many directions. They show, among others, anticancer activity in studies on many cell lines, e.g., breast cancer cells (MCF-7 [1,2,3,4], MDA-MB-231 [5], BT-474 [3] and MDA435 [6]), lung cancer (NCI-H460 [4] and H460a [6]), central nervous system CNS cancer (SF-268) [4], colon cancer (HCT116, SW480, RKO), and osteosarcoma (SJSA1) [6]. Their anticancer activity is related, among others, to the inhibition of the activity of certain enzymes, e.g., cyclin-dependent kinase 1 [6], cyclin-dependent kinase 2 (CDK2) [1], carbonic anhydrase IX (CA IX) [5], and human mitotic kinesin Eg5 [7].

Pseudothiohydantoin derivatives exhibit antibacterial activity, both on Gram-positive (*Bacillus subtilis*, *Staphylococcus aureus*, *Corynebacterium diphtheriae*, and *Enterococcus faecalis*) and Gram-negative (*Escherichia coli*, *Pseudomonas aeruginosa*, *Klebsiella pneumonia*, *Proteus vulgaris*, and *Salmonella typhimurium*) bacteria [3,5,8,9,10]. Piperazinyl–tetrazole derivatives of thiazolone have activity against *Mycobacterium tuberculosis* and have been considered as potential antitubercular agents [11]. The antifungal activity of 2-aminothiazolone derivatives was also tested against the following fungal strains: *Aspergillus oryzae*, *Penicillium chrysogenum*, *Fusarium oxysporum*, *Candida albicans*, *Aspergillus flavus* and *Aspergillus niger* [3,5]. Compounds containing thiazolone moiety have also been shown to be HCV NS5B polymerase allosteric inhibitors, thereby making them potential anti-hepatitis C virus drugs [12,13].

Pseudothiohydantoin derivatives are also an interesting and promising class of compounds in the search for selective 11β-hydroxysteroid dehydrogenase type 1 (11β-HSD1) inhibitors [14,15,16,17,18,19]. Pseudothiohydantoin is an enzyme that catalyzes the conversion of inactive cortisone to active cortisol. The inhibition of the activity of 11β-hydroxysteroid dehydrogenase type 1 reduces the level of cortisol, and thus reduces the mass of adipose tissue, insulin resistance, and central obesity, as well as lowers the level of total cholesterol. Selective inhibitors have significant potential as a pharmacological treatment for type 2 diabetes, obesity, and cardiovascular disease [20,21].

One of the requirements for the use of a substance as a medicine is its bioavailability. The assessment of the usefulness of a substance as an orally administered drug based on its properties is defined by Lipinski’s rule of five. This rule was formulated on the basis of an analysis of the physicochemical properties of compounds that have already been registered as oral drugs in terms of four parameters that have a key impact on the absorption and penetration of chemical compounds. One of them is lipophilicity [22]. This parameter, is expressed as a partition coefficient P or its decimal logarithm (log P). After entering the body, the drug must overcome a number of barriers in the form of biological membranes to reach its place of action. Drugs are transported across cell membranes by the so-called passive diffusion. Nonionized drug molecules dissolved in the aqueous phase pass through the semipermeable lipid membrane into the aqueous phase on the other side of the membrane. The speed of this process depends, among others, on the solubility of the drugs in the lipids, which is determined by lipophilicity. Lipinski’s rule of five describes the molecular properties that are important for drug pharmacokinetics in the human body says, and that value for an oral active drug is log P ≤ 5 [22].

In the development of new drugs, in addition to predicting physicochemical parameters, an important step is also the determination and evaluation of ADME properties (absorption, distribution, metabolism, and excretion). This is because ADME’s poor properties, beside inefficiency and toxicity, are still a significant cause of failure and high cost in drug design [23]. Predicting ADME using in silico methods allows for eliminating compounds with undesirable ADME properties in order to minimize the risk of drug failure related to pharmacokinetics in later phases of clinical trials [24]. In addition, ADME parameters provide initial information on the dosing schedule and the size of the drug dose.

In the present work, we evaluated the chromatographic lipophilicity parameters of 28 pseudothiohydantoin derivatives showing inhibitory activity towards 11β-HSD1 using RP-TLC and RP-HPLC methods, and we compared them with calculated log P values. In addition, we determined and evaluated the ADME parameters of the compounds in order to appropriately select potential drug candidates.

The series of pseudothiohydantoin derivatives were synthesized via the reaction of *N*-alkylthiourea with a corresponding 2-bromo ester. The reactions were carried out in various solvents and conditions, and their detailed description is provided in refs. [25,26,27].

## 2. Results and Discussion

### 2.1. Lipophilicity Studies

The lipophilicity parameters log k_w_ and R_M0_ were determined for 28 2-aminothiazol-4(5*H*)-one derivatives, which were divided into three groups that differed in the substituent in the amino group (Figure 1). These derivatives included 2-isopropylaminothiazol-4(5*H*)-one (compounds: **1**–**9**), 2-*tert*-butylaminothiazol-4(5*H*)-one (compounds: **10**–**18**), and 2-adamantylaminothiazol-4(5*H*)-one (compounds: **19**–**28**).

In both methods, methanol was chosen as the most recommended organic modifier of the mobile phases for lipophilicity estimation by RP-TLC and RP-HPLC, since it does not disturb the hydrogen bonding network of water [28,29].

The linear relationship between log k_w_ or R_M0_ and the methanol concentration was determined on the basis of Formulas (2) and (5). For all three groups of derivatives, in a wide range of organic modifier concentrations in the mobile phase, high values of correlation coefficients (r = 0.92 − 0.99 for RP-HPLC and r = 0.82 − 0.97 for RP-TLC) made it possible to determine the lipophilicity parameters R_M0_ and log k_w_ by extrapolation (Table 1).

It was observed that, for all compounds, the log k_w_ values determined by RP-HPLC (1.35–5.63) were higher than the R_M0_ values determined by RP-TLC (0.94–3.56) (Figure 2).

The results obtained for both methods indicate a relationship between the structure of the tested compounds and the lipophilicity parameters (Table 1). The analysis of the values of the lipophilicity parameters determined by RP-TLC and RP-HPLC showed their dependence on the substituents present in the 2- and 5-position of the thiazole ring. For most of the tested compounds, it was found that, the larger the substituent in the 2-position, the higher the log k_w_ and R_M0_ values, while the same groups were maintained in the C-5 position (Figure 3 and Figure 4).

The lowest values of log k_w_ and R_M0_ were determined for the compound containing the smallest volume substituents at C-5 and in the amino group at C-2, as well as 5-methyl-2-isopropylaminothiazol-4(5*H*)-one (**1**), and they were 1.34 and 0.94, respectively. The presence of larger substituents in the 2-position of the thiazole ring, i.e., the *tert*-butylamino and adamantylamino group, while maintaining the methyl group in the 5-position (**10**, **19**) caused an increase in both lipophilicity parameters.

A similar relationship was observed in the case of derivatives containing other aliphatic substituents at C-5, i.e., a ethyl (compounds **2**, **11**, **20**), propyl (compounds **3**, **12**, **21**), isopropyl (compounds **4**, **13**, **22**), phenyl (compounds **6**, **15**, **24**), 4-bromophenyl (compounds **7**, **16**, **25**), and a cyclohexane ring in spiro connection to the thiazolone ring (compounds **8**, **17**, **26**). When the cyclohexane ring was replaced by a 4-membered ring, higher values of both the log k_w_ and R_M0_ were observed for the isopropylamino derivative (**8**) than for the *tert*-butylamino derivative (**17**). An analogous situation was observed for the R_M0_ parameter in the case of compounds containing two methyl groups in the 5-position (compounds **5** and **14**).

For all three groups of derivatives, it was found that, with the increase in the chain length of the substituent in the 5-position of the thiazole ring, the log k_w_ value increased. For ethyl derivatives (**2**, **11**, **20**) the log k_w_ was respectively: 1.86, 1.88, and 3.6. A similar dependence took place for the R_M0_ parameter (values were respectively: 1.39, 1.59, and 2.30). For comparison, the results of both parameters for propyl derivatives (**3**, **12**, **21**) were higher and amounted to the following: log k_w_—2.28, 2.46, and 4.08 andR_M0_—1.63, 1.71, and 2.33, respectively.

In the presence of substituents with the same number of carbon atoms in the 5-position of the thiazole ring, it was observed that a molecule containing a more branched substituent was characterized by lower values of the lipophilicity parameters log k_w_ and R_M0_. Such a relationship was found for compounds with a propyl (**3**, **12**, **21**) and isopropyl group (**4**, **13**, **22**). The exceptions were the equal log k_w_ values for compounds **3** and **4**.

For derivatives with spiro systems of thiazole and alicyclic rings, in accordance with previous observations, it was found that compounds with a smaller structure (**9**, **18**, **27**) had lower lipophilicity parameters compared to more complex derivatives (**8**, **26**). An exception was 2-(*tert*-butylamino)-1-thia-3-azaspiro[4.5]dec-2-en-4-one (**17**), which showed lower log k_w_ and R_M0_ values than compound **8**, even though the C-2 position had a larger substituent (*tert*-butyl group).

In the case of the presence of a phenyl substituent at the C-5 of the thiazole moiety in the molecule, a clear increase in the log k_w_ parameters (from 2.84 for **6** to 4.13 for **24**) and R_M0_ parameters (from 1.58 for **6** to 3.31 for **24**) was observed for all three groups of compounds compared to the alkyl derivatives. The presence of a bromine atom in the 4-position of the phenyl ring resulted in an even greater increase in the lipophilicity parameters. The highest values of the log k_w_ and R_M0_ were observed for the derivative with the most extensive structure, i.e., for 2-(adamantylamino)-5-(4-bromophenyl)thiazol-4(5*H*)-one (**25**), which were 5.63 and 3.56, respectively.

2-(Adamantylamino)thiazol-4(5*H*)-one (**28**) lacked a substituent in the 5-position of the thiazole system and had the lowest log k_w_ and R_M0_ values of all adamantyl derivatives. However, these values were higher compared to the isopropyl and *tert*-butyl derivatives containing alkyl substituents in the 5-position, which was influenced by the presence of a large hydrophobic adamantyl substituent.

The experimentally obtained log k_w_ and R_M0_ parameters were compared with the values of lipophilicity determined by theoretical methods using the VCCLAB calculation program [30] (Table 2).

Based on the analysis of the obtained data, it can be concluded that, in the case of nine compounds (**7**, **9**–**11**, **14**, **16**, **18**, **20**, and **23**) the MLOGP values were the closest to the experimentally determined log k_w_ parameters (Figure 5, Figure 6 and Figure 7). For eight derivatives (**1**, **3**, **4**, **6**, **12**, **13**, **19**, and **26**) the best correspondence of the log k_w_ with the parameter XLOGP3 was observed. Eight other derivatives (**5**, **8**, **15**, **22**, **25**, **27**, and **28**) had log k_w_ values close to the calculated ALOGP, three compounds (**2**, **23**, and **24**) were close to the AlogPs, and two compounds (**17** and **21**) were close to the milogP. The calculated AClogP and XLOGP2 parameters deviated the most from the experimental log k_w_ values.

On the other hand, in the case of the experimentally determined R_M0_ parameters, for as many as 17 derivatives (**1**, **2**, **8**, **9**, **12**–**14**, **17**, and **20**–**27**) they were most similar to the AClogP parameter. In the case of seven derivatives (**3**, **4**, **7**, **11**, **16**, **19**, and **28**), the convergence of the R_M0_ value with the XLOGP2 parameter was observed. Derivatives **5**, **6**, and **18** had the R_M0_ values closest to the calculated miLogP parameters. For the remaining three compounds, the parameter R_M0_ was nearest to the calculated value: the AlogPs for compound **7**, the ALOGP for compound **15**, and the MLOGP for compound **10**.

The above data show that it is difficult to indicate such a calculation parameter that would be the most optimal for the discussed group of compounds. In addition, it is difficult to find a relationship between the adjustment of the computational parameters and the structure of the analyzed compounds. The varied values of calculation and experimental lipophilicity parameters show that the experimental determination of lipophilicity parameters for the group of tested compounds was necessary and fully justified.

### 2.2. In Silico ADME Prediction

Physicochemical properties allowing for the assessment as to whether compounds **1**–**28** meet Lipinski’s rule of five and Veber’s rules, which have been described in our previous studies [25,26,27]. The analysis of the results showed that all the analyzed compounds met the criteria of the rule of five and Veber’s rule, which proves their good oral bioavailability. Currently, we determined and evaluated the ADME parameters of the compounds in order to appropriately select potential drug candidates.

Analysis of the ADME properties were performed using the pkCSM server [31], which predicts the pharmacokinetic properties of small molecules [32]. The pharmacokinetic results of the compounds are presented in Table 3.

#### 2.2.1. Absorption

The absorption of pseudothiohydantoin derivatives was assessed using factors such as water solubility, Caco-2 cell permeability (human colon adenoma cells), and human intestinal absorption. Water solubility was measured using the logS parameter (S—solubility expressed in mol/L). All of the 28 compounds tested were characterized by low solubility in water due to the large number of carbon atoms and the lack of polar, hydrophilic moieties. Compound **1** had the lowest molecular weight, wherein it contained an isopropylamine substituent in the 2-position and a methyl group in the 5-position of the thiazolone ring, which had the highest solubility in water (logS = 1.477). In contrast, compound **25** had the highest molecular weight, which contained two hydrophobic moieties on the thiazolone ring (adamantylamine substituent in position 2 and 4-bromophenyl substituent in position 5), which had the lowest solubility with a log S = −5.394.

Due to the functional and morphological similarity of Caco-2 cells to the human intestinal epithelium, the study of the permeability of compounds through the monolayer of Caco-2 cells is the most commonly used in vitro method to determine the absorption of orally administered drugs [33]. The predicted Caco-2 permeability for the tested compounds is given as the logarithm of the apparent permeability coefficient (log Papp). Compounds having a predicted log Papp in 10^−6^ cm/s greater than 0.9 are considered to have high Caco-2 permeability. According to the results of the analysis, all the synthesized compounds showed a high permeability of Caco-2 cells (values ranged from 1.08 to 1.614).

Calculated human intestinal absorption values showed that all compounds had an excellent probability of intestinal absorption (values range from 90.77% to 95.75%). The 2-Aminothiazol-4(5*H*)-one derivatives containing a 4-bromophenyl substituent at the C-5 position of the thiazole ring were characterized by the lowest parameters of intestinal absorption, regardless of the type of substituent at the C-2.

#### 2.2.2. Distribution

Distribution was predicted using descriptors such as the blood–brain permeability (BBB), steady-state volume of distribution (VDss), and unbound fraction (FU). The blood–brain barrier is a structure that plays an important role in maintaining brain homeostasis. The BBB controls the transfer of essential substances needed for the proper functioning of the brain [34]. In addition, it takes part in the removal of cellular metabolites and toxins [35]. Predicting this value is critical in drug discovery, because compounds must first cross the blood–brain barrier in order to work in the brain. The predicted BBB membrane permeability (log BB) values of our compounds ranged from −0.235 to 0.246, which means that they will pass through the BBB membrane.

VDss is an important pharmacokinetic parameter that, together with clearance, determines the half-life and, thus, influences the dosing regimen [36]. This parameter represents the degree of distribution of the drug in tissues but not in plasma. The predicted log VDss values for most of the tested derivatives ranged from -0.15 to 0.45, which means that the compounds had an optimal constant distribution in the plasma and tissues. Two compounds, 2-(adamantan-1-ylamino)-5-phenylthiazol-4(5*H*)-one (**24**) and 2-(adamantan-1-ylamino)-5-(4-bromophenyl)thiazol-4(5*H*)-one (**25**), possessed suboptimal values >0.45, meaning that most of the compound will be bound in tissues instead of being bound to plasma.

In addition to the VDss, the plasma unbound fraction (FU) is an important pharmacokinetic parameter. This parameter determines the amount of the drug that is “free” in the plasma, and thus the fraction that is able to diffuse from the plasma into the tissues and interact with pharmacologically active proteins [37]. In addition, the FU affects glomerular filtration and hepatic metabolism [37]. In the current work, it was predicted that the plasma unbound fraction for all 2-aminothiazol-4(5*H*)-one derivatives would have low values in the range of 0.033 and 0.606.

#### 2.2.3. Metabolism

The metabolism of the tested derivatives was assessed in terms of their inhibitory effect on the five main cytochrome P450 enzymes, namely, CYP2D6, CYP3A4, CYP1A2, CYP2C19, and CYP2C9 (these enzymes are responsible for the metabolism of almost 80% of drugs) [38]. The inhibition of the CYP family of enzymes affects the biotransformation and clearance of the drug, which may lead to an increase in its plasma concentration [39]. CYP inhibition therefore leads to increased toxicity or a lack of therapeutic effect of the drug [39]. CYP3A4 is responsible for the metabolism of 50% of all drugs, which makes it the most important cytochrome P450 isoform [40]. CYP2D6 is responsible for the metabolism of 25% of the drugs available on the market and is highly polymorphic [40]. The genetic polymorphism of this enzyme is related to the fact that some people may have increased or decreased activity of this enzyme, which may lead to side effects or reduce the effectiveness of the drug [41]. On the other hand, CYP2C9 is mainly responsible for the metabolism of drugs with a narrow therapeutic index [42]. The CYP1A2 enzyme is involved in the metabolism of about 9% of marketed drugs, including drugs with a narrow therapeutic index, including theophylline [43]. CYP2C19, in addition to metabolizing several drugs, is also involved in cholesterol metabolism and the physiology of steroid hormones [44]. Nine compounds **7**, **15**–**16, 20**–**22**, and **24**–**26** showed an inhibitory effect on CYP2C19 (Table 4). Only 2-adamantylaminothiazol-4(5*H*)-one derivatives containing an aromatic system in the C-5 position of the thiazole ring (compounds **24**–**25**) turned out to be inhibitors of CYP2D6. In addition, molecule **6**–**7** and **15**–**16** were predicted to possibly inhibit CYP1A2. Among the tested derivatives, only compound **16** was found to be an inhibitor of CYP2C9. All compounds obtained will not inhibit CYP3A4.

#### 2.2.4. Excretion

In the case of the elimination process, one of the main parameters describing this process is clearance (CL). CL is the volume of plasma that is cleared of substances, including drugs, per unit of time. The parameter in terms of which the process of excretion of the tested compounds was assessed was the total/total clearance (CL_tot_). The total clearance determines the efficiency of drug elimination from the whole body, without indicating specific mechanisms of elimination [24]. CL_tot_ is a useful parameter, among others, to determine the dosing rate [45]. Among the pseudothiohydantoin derivatives tested, the CL_tot_ values were low and ranged from -0.575 to 0.227 mL/min/kg, which thus are likely to result in longer intervals between individual doses.

ADME analysis showed adequate absorption, distribution, and elimination parameters for most compounds; therefore, good bioavailability levels are expected for pseudothiohydanotin derivatives. In the case of metabolism, none of the tested derivatives turned out to be an inhibitor of the most important CYP3A4 isoenzyme, so they will not cause changes to the actions of other drugs metabolized by this enzyme. Together, these results showed that the tested compounds are good potential drug candidates.

## 3. Materials and Methods

The compounds studied in this work have been synthesized earlier, and their synthesis and full characterization are described in [25,26,27].

### 3.1. Determination of Lipophilicity Parameters Using the Reversed-Phase HPLC Technique [46,47]

The HPLC experiments were performed on the Shimadzu HPLC system (Kyoto, Japan) equipped with solvent delivery pump LC-20AD, UV–VIS detector model SPD-20A, degasser model DGU-20A5, a column oven model CTO-20A, and a column Superspher^®^ 100 RP-18 (4 μm), Merck (Darmstadt, Germany). Methanol and water for HPLC from POCH (Gliwice, Poland) were used as solvents. Mobile phase methanol/water varied in the various ratios. Methanol concentration expressed as volume fraction *v*/*v*, varied in the range from 0.55 to 0.95 in constant steps of 0.05. Compounds **1**–**28** were dissolved in methanol (1 mg/mL). The mobile phase flow rate was 0.7 mL/min. The injection volume was 0.02 mL. All analyzes were performed at 25 °C. Experiments were performed in triplicate with similar results.

The retention coefficient k was calculated from the following relationship:(1)$$k=\frac{t_{r}}{t_{m}}-1,$$
where t_r_ [min]—retention time, and t_m_ [min]—time for dead volume (measured by use of uracil).

To determine the linear relationship between the values of the log k parameter and the concentration of methanol in the mobile phase, the Soczewiński–Wachtmeister equation was used:(2)$$\mathrm{logk}={\;\log\;k}_{w}+S\Phi ,$$
where $\log\;k_{w}$—value extrapolated to zero methanol concentration, Φ—methanol concentration in the mobile phase (volume fraction *v*/*v*), and S—the slope of the regression curve.

The parameter $\Phi_{0}$ was calculated from the equation:(3)$$\Phi_{0}=-\frac{{\log\;k}_{w}}{S}.$$

### 3.2. Determination of Lipophilicity Parameters Using the Reversed-Phase TLC Technique [46]

Specifically, 10 × 10 cm plates with silica gel HPTLC 60RP-18 WF254s (Merck, Darmstadt, Niemcy) were used as the stationary phase. Methanol and water for HPLC from POCH (Gliwice, Poland) were used as solvents. Methanol concentration expressed as volume fraction *v*/*v*, varied in the range from 0.5 to 0.7 in constant steps of 0.05. Test compounds were dissolved in methanol (2 mg/mL), and then samples (0.01 mL) of each were applied to plates, which were developed in a horizontal DS chamber (Chromdes, Lublin, Poland).

The developing distance was 8 cm. Developed plates were air-dried and observed under 254 nm ultraviolet lamp. All analyzes were performed at 22 °C. Experiments were performed in triplicate with similar results.

For all the compounds, the relative lipophilicity R_M_ values for five methanol–water mobile phases were calculated by the use of the following formula:(4)$$R_{M}=\log\left[\frac{1-R_{F}}{R_{F}}\right].$$

The linear relationship between the R_M_ values and the concentration of methanol in the mobile phase to determine the lipophilicity parameters R_M0_ is described by the following equation:(5)$$R_{M}=RM_{0}+S\Phi ,$$
where Φ—methanol concentration in the mobile phase (volume fraction *v*/*v*), and S—the slope of the regression curve.

The quantity $\Phi_{0}$ was calculated from the following equation:(6)$$\Phi_{0}=-\frac{R_{M0}}{S}.$$

## 4. Conclusions

In this manuscript, the lipophilicity parameters of twenty eight 2-aminothiazol-4(5*H*)-one derivatives were determined by RP-HPLC (log k_w_) and RP-TLC (R_M0_). For all tested compounds, a linear correlation was obtained between the log k_w_ and R_M0_ values and the concentration of the organic modifier in the mobile phase (methanol). In the case of the analyzed compounds, the RP-HPLC method turned out to be more exact, which resulted from the comparison of the correlation coefficients. For all analyzed compounds, it was observed that the log k_w_ values were higher than the R_M0_ values. Comparison of the experimentally obtained log k_w_ and R_M0_ parameters with the lipophilicity values determined by theoretical methods showed that it is difficult to fit the calculation parameter that would be the most optimal for the discussed group of compounds. In addition, it is difficult to find a relationship between the adjustment of the computational parameters and the structure of the analyzed compounds. Such varied values show that the experimental determination of lipophilicity parameters for the group of tested compounds was necessary and fully justified.

For most of the tested compounds, both experimentally and computationally determined lipophilicity parameters were lower than five, so they met the Lipinski rule. Therefore, they may be orally active when used as drugs. An exception was 2-(adamantylamino)-5-(4-bromophenyl)thiazol-4(5*H*)-one (**25**).

ADME analysis showed adequate absorption, distribution, and elimination parameters for most compounds; therefore, good bioavailability levels are expected for pseudothiohydanotin derivatives. None of the tested derivatives turned out to be an inhibitor of the most important CYP3A4 isoenzyme, so they will not cause changes to the action of other drugs metabolized by this enzyme.

Both the determined lipophilicity parameters and the ADME analysis showed that the tested compounds are good potential drug candidates.

## Figures and Tables

**Figure 1 ijms-24-12230-f001:**
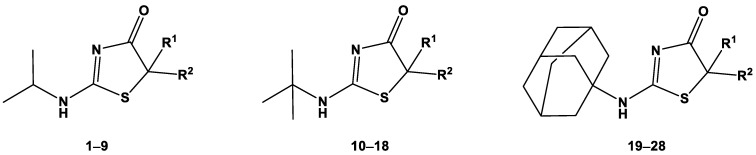
Structures of the tested compounds.

**Figure 2 ijms-24-12230-f002:**
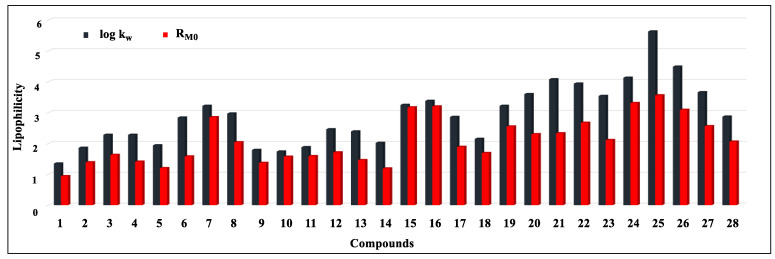
Comparison of R_M0_ and log k_w_ values experimentally determined for compounds **1**–**28**.

**Figure 3 ijms-24-12230-f003:**
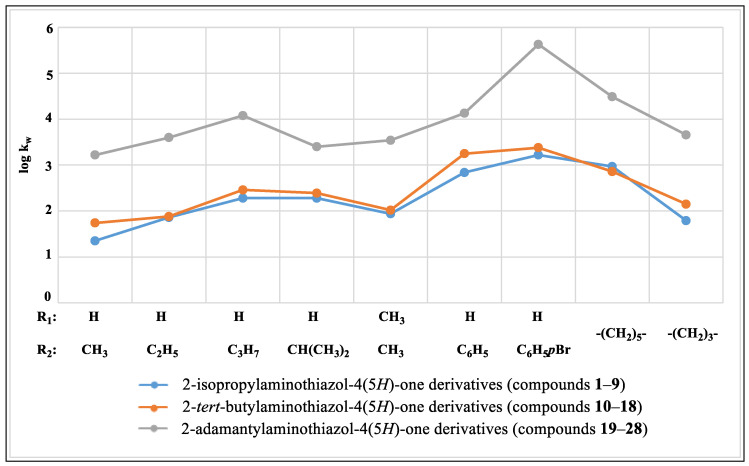
The relationship of log k_w_ to substituents in the 5-position.

**Figure 4 ijms-24-12230-f004:**
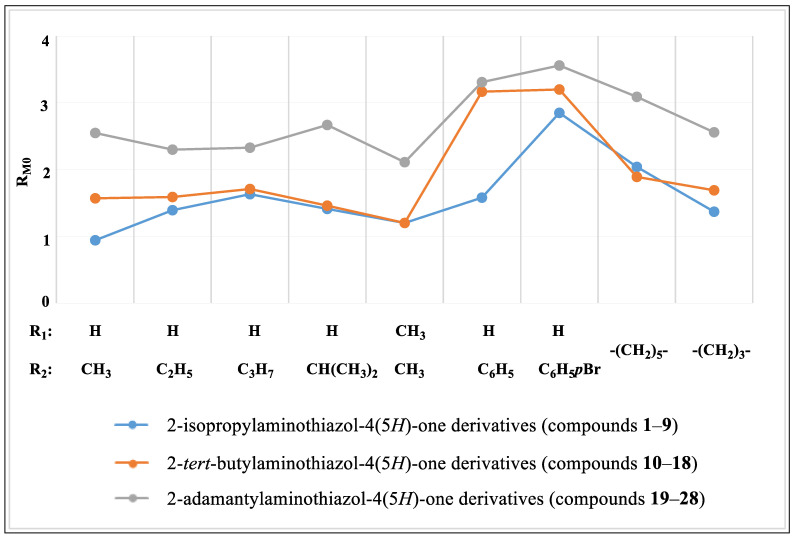
The relationship of R_M0_ to substituents in the 5-position.

**Figure 5 ijms-24-12230-f005:**
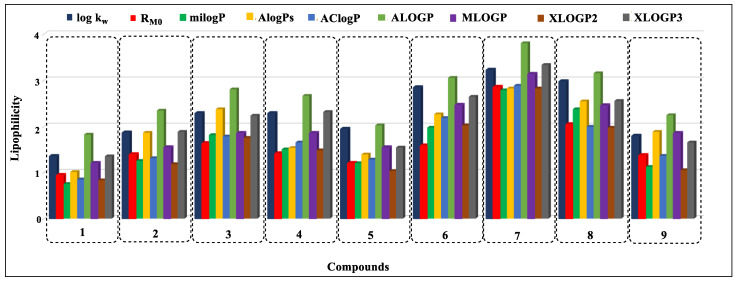
Comparison of lipophilicity parameters obtained by experimental and calculation methods for compounds **1**–**9**.

**Figure 6 ijms-24-12230-f006:**
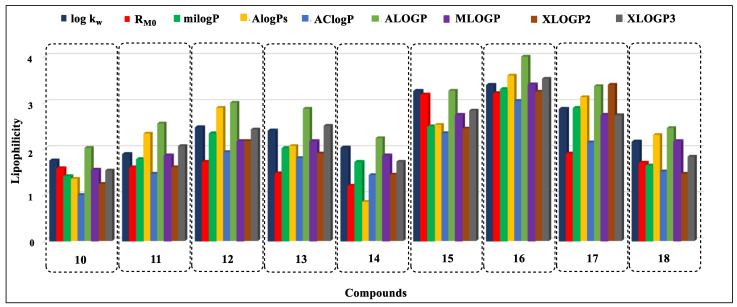
Comparison of lipophilicity parameters obtained by experimental and calculation methods for compounds **10**–**18**.

**Figure 7 ijms-24-12230-f007:**
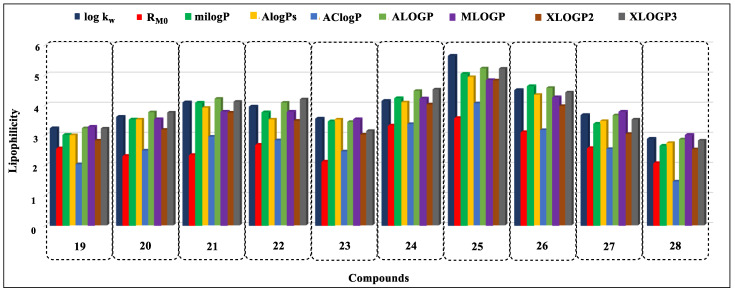
Comparison of lipophilicity parameters obtained by experimental and calculation methods for compounds **19**–**28**.

**Table 1 ijms-24-12230-t001:** The lipophilicity parameters calculated from RP-TLC and RP-HPLC experimental values.

No.	R^1^	R^2^	RP–HPLC				RP–TLC			
			log k_w_	−S	Φ_0_	R	R_M0_	−S	Φ_0_	R
**1**	H	CH_3_	1.35	2.975	0.454	0.9953	0.94	1.280	0.734	0.8623
**2**	H	C_2_H_5_	1.86	3.254	0.571	0.9885	1.39	2.075	0.668	0.9590
**3**	H	C_3_H_7_	2.28	3.494	0.652	0.9988	1.63	2.422	0.673	0.9627
**4**	H	CH(CH_3_)_2_	2.28	3.603	0.633	0.9942	1.41	2.127	0.663	0.9462
**5**	CH_3_	CH_3_	1.94	3.400	0.571	0.9855	1.20	1.903	0.629	0.9367
**6**	H	C_6_H_5_	2.84	4.3069	0.659	0.9767	1.58	2.049	0.771	0.9619
**7**	H	C_6_H_4_*p*-Br	3.22	4.422	0.728	0.9956	2.85	3.870	0.736	0.9689
**8**	C_5_H_10_		2.97	4.170	0.712	0.9652	2.04	2.877	0.709	0.9530
**9**	C_3_H_6_		1.79	2.890	0.619	0.9958	1.37	1.990	0.691	0.9789
**10**	H	CH_3_	1.74	3.122	0.557	0.9901	1.57	2.393	0.658	0.9572
**11**	H	C_2_H_5_	1.88	3.073	0.612	0.9927	1.59	2.478	0.643	0.9686
**12**	H	C_3_H_7_	2.46	3.606	0.682	0.9979	1.71	2.426	0.703	0.9325
**13**	H	CH(CH_3_)_2_	2.39	3.585	0.667	0.9974	1.46	2.095	0.699	0.9075
**14**	CH_3_	CH_3_	2.02	3.320	0.608	0.9932	1.19	1.826	0.652	0.9207
**15**	H	C_6_H_5_	3.25	4.449	0.730	0.9623	3.17	4.293	0.739	0.9371
**16**	H	C_6_H_4_*p*-Br	3.38	4.537	0.745	0.9984	3.20	4.131	0.774	0.9662
**17**	C_5_H_10_		2.86	3.904	0.732	0.9993	1.89	2.477	0.762	0.8269
**18**	C_3_H_6_		2.15	3.297	0.652	0.9987	1.69	2.476	0.683	0.8583
**19**	H	CH_3_	3.22	4.099	0.786	0.9990	2.55	3.148	0.810	0.9107
**20**	H	C_2_H_5_	3.60	4.437	0.811	0.9988	2.30	2.678	0.860	0.9657
**21**	H	C_3_H_7_	4.08	4.859	0.840	0.9983	2.33	2.564	0.909	0.9001
**22**	H	CH(CH_3_)_2_	3.94	4.744	0.830	0.9984	2.67	3.175	0.872	0.8719
**23**	CH_3_	CH_3_	3.54	4.392	0.806	0.9989	2.11	2.446	0.863	0.9127
**24**	H	C_6_H_5_	4.13	5.024	0.822	0.9984	3.31	4.203	0.787	0.9636
**25**	H	C_6_H_4_*p*-Br	5.63	5.578	1.009	0.9291	3.56	3.277	1.086	0.9226
**26**	C_5_H_10_		4.49	5.192	0.865	0.9974	3.09	3.520	0.879	0.9624
**27**	C_3_H_6_		3.66	4.383	0.835	0.9973	2.56	3.025	0.847	0.9797
**28**	H	H	2.87	3.783	0.759	0.9993	2.06	2.482	0.832	0.9770

RP-HPLC—reversed phase high-performance liquid chromatography; RP-TLC—reversed phase thin layer chromatography; log kw—decimal logarithm (log P) of partition coefficient P; S—slope of the regression curve; Φ_0_—chromatographic lipophilicity parameter; R—correlation coefficient; R_M0_—value extrapolated to zero methanol concentration.

**Table 2 ijms-24-12230-t002:** Values of lipophilicity parameters calculated and determined experimentally.

No.	Log k_w_	R_M0_	milogP	AlogPs	AClogP	ALOGP	MLOGP	XLOGP2	XLOGP3
**1**	1.35	0.94	0.75	1.01	0.85	1.82	1.20	0.83	1.35
**2**	1.86	1.39	1.25	1.86	1.31	2.34	1.54	1.18	1.88
**3**	2.28	1.63	1.81	2.37	1.78	2.80	1.85	1.75	2.23
**4**	2.28	1.41	1.50	1.53	1.65	2.66	1.85	1.48	2.31
**5**	1.94	1.20	1.20	1.39	1.28	2.02	1.54	1.03	1.54
**6**	2.84	1.58	1.97	2.26	2.18	3.05	2.46	2.02	2.64
**7**	3.22	2.85	2.78	2.82	2.88	3.80	3.13	2.82	3.33
**8**	2.97	2.04	2.37	2.54	1.99	3.15	2.45	1.97	2.55
**9**	1.79	1.37	1.12	1.88	1.36	2.24	1.85	1.05	1.65
**10**	1.74	1.57	1.40	1.35	1.00	2.02	1.54	1.24	1.53
**11**	1.88	1.59	1.77	2.33	1.46	2.55	1.85	1.60	2.06
**12**	2.46	1.71	2.33	2.89	1.93	3.00	2.16	2.17	2.42
**13**	2.39	1.46	2.01	2.06	1.80	2.87	2.16	1.90	2.50
**14**	2.02	1.19	1.71	0.85	1.43	2.23	1.85	1.44	1.72
**15**	3.25	3.17	2.48	2.52	2.34	3.26	2.73	2.44	2.83
**16**	3.38	3.20	3.29	3.59	3.04	4.01	3.39	3.24	3.52
**17**	2.86	1.89	2.88	3.12	2.14	3.36	2.73	3.39	2.73
**18**	2.15	1.69	1.63	2.30	1.51	2.45	2.16	1.46	1.83
**19**	3.22	2.55	3.00	3.00	2.03	3.23	3.27	2.82	3.22
**20**	3.60	2.30	3.51	3.52	2.49	3.76	3.52	3.18	3.75
**21**	4.08	2.33	4.07	3.91	2.95	4.21	3.77	3.75	4.11
**22**	3.94	2.67	3. 75	3.52	2.83	4.08	3.77	3.48	4.19
**23**	3.54	2.11	3.45	3.52	2.46	3.44	3.52	3.02	3.14
**24**	4.13	3.31	4.22	4.09	3.37	4.47	4.21	4.02	4.52
**25**	5.63	3.56	5.03	4.93	4.06	5.22	4.82	4.82	5.21
**26**	4.49	3.09	4.62	4.34	3.17	4.57	4.25	3.97	4.42
**27**	3.66	2.56	3.37	3.47	2.54	3.66	3.77	3.04	3.52
**28**	2.87	2.06	2.64	2.74	1.46	2.86	3.00	2.53	2.82

**Table 3 ijms-24-12230-t003:** Calculation of ADME properties (absorption and distribution) of the compounds **1**–**28** using pkcSM software [31].

Compound	Absorption			Distribution		
	Water Solubility (log mol/L)	Caco-2 Premeability (log Papp in 10^−6^ cm/s)	Intestinal Absorbtion (%abs)	VDss (Log L/kg)	BBB Premeability (log BB)	Fract. Unb. (FU)
**1**	−1.477	1.393	95.751	−0.102	−0.235	0.606
**2**	−2.014	1.484	93.679	−0.074	−0.191	0.516
**3**	−2.459	1.483	92.838	−0.03	0.055	0.466
**4**	−2.198	1.113	94.017	−0.098	0.05	0.472
**5**	−1.831	1.395	94.12	−0.079	−0.219	0.577
**6**	−3.402	1.424	93.276	0.147	0.123	0.201
**7**	−4.349	1.227	91.821	0.133	0.078	0.184
**8**	−3.206	1.08	93.185	0.032	0.063	0.341
**9**	−2.386	1.106	94.229	0.012	−0.173	0.47
**10**	−2.124	1.495	93.305	−0.137	−0.208	0.49
**11**	−2.519	1.495	92.685	−0.097	0.109	0.442
**12**	−2.946	1.494	91.845	−0.057	0.097	0.394
**13**	−2.976	1.51	92.211	−0.094	0.107	0.375
**14**	−2.311	1.407	93.34	−0.077	0.173	0.518
**15**	−3.968	1.437	92.51	0.144	0.151	0.145
**16**	−4.893	1.239	91.055	0.129	0.106	0.128
**17**	−3.673	1.093	92.419	0.006	0.09	0.282
**18**	−2.866	1.118	93.462	−0.001	0.092	0.409
**19**	−4.228	1.457	93.571	0.42	0.233	0.326
**20**	−4.43	1.456	92.938	0.435	0.219	0.281
**21**	−4.635	1.455	92.091	0.448	0.208	0.234
**22**	−4.693	1.471	92.45	0.404	0.235	0.218
**23**	−4.477	1.459	93.182	0.437	0.246	0.297
**24**	−4.889	1.614	92.226	0.677	0.226	0.079
**25**	−5.394	1.154	90.777	0.641	0.197	0.033
**26**	−5.026	1.468	91.916	0.36	0.218	0.118
**27**	−4.753	1.457	92.783	0.429	0.169	0.205
**28**	−3.943	1.445	93.949	0.421	0.222	0.376

**Table 4 ijms-24-12230-t004:** Calculation of ADME properties (metabolism and excretion) of the compounds **1**–**28** using pkcSM software [31].

Compound	Metabolism					Excretion
	CYP2D6	CYP3A4	CYP1A2	CYP2C19	CYP2C9	Total Clearance (log (mL/min/kg))
	I	I	I	I	I	
**1**	No	No	No	No	No	0.164
**2**	No	No	No	No	No	0.203
**3**	No	No	No	No	No	0.227
**4**	No	No	No	No	No	0.187
**5**	No	No	No	No	No	0.091
**6**	No	No	Yes	No	No	0.023
**7**	No	No	Yes	Yes	No	−0.129
**8**	No	No	No	No	No	0.043
**9**	No	No	No	No	No	−0.043
**10**	No	No	No	No	No	0.061
**11**	No	No	No	No	No	0.1
**12**	No	No	No	No	No	0.123
**13**	No	No	No	No	No	0.083
**14**	No	No	No	No	No	−0.013
**15**	No	No	Yes	Yes	No	−0.083
**16**	No	No	Yes	Yes	Yes	−0.235
**17**	No	No	No	No	No	−0.063
**18**	No	No	No	No	No	−0.149
**19**	No	No	No	No	No	−0.302
**20**	No	No	No	Yes	No	−0.264
**21**	No	No	No	Yes	No	−0.239
**22**	No	No	No	Yes	No	−0.283
**23**	No	No	No	No	No	−0.381
**24**	Yes	No	No	Yes	No	−0.422
**25**	Yes	No	No	Yes	No	−0.575
**26**	No	No	No	Yes	No	−0.403
**27**	No	No	No	No	No	−0.488
**28**	No	No	No	No	No	−0.24

## Data Availability

Data from the research described in the manuscript are available from the authors.

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
