# Peer review of "Lipophilic Studies and In Silico ADME Profiling of Biologically Active 2-Aminothiazol-4(5H)-one Derivatives"

_ijms, 2023, doi:10.3390/ijms241512230_

Round 1
Reviewer 1 Report
The overall presentation of the manuscript is correct, well written, and the references are updated. Since the authors only determined experimentally the log kw of the compounds, rather than discussing predictive ADME properties in so much detail, it is essential to include and discuss the experimental determination of other physicochemical properties that will substantiate with more confidence both the Lispinki rule and predictive ADME properties.
Minor issues
Abstract: line 12 – please change to “biological activities”
Authors should put the bacteria and fungi species names in italics.
Introduction lines 79-80: Authors said: “...because poor ADME properties are the main reason for failures and high costs in the design of new drugs [23]”. For many years this statement was true. However, currently, it no longer corresponds to reality. Currently, the main causes for failures are lack of efficacy and toxicity. Authors should update this statement and include a more recent reference.
Results and Discussion: line 130. It is not clear what the authors mean by “…compound with the least developed structure.” To avoid misunderstandings, this sentence should be changed.
Materials and Methods: Authors should add, in both chromatographic methods, the number of tests that were performed for each of the mobile phases.
Author Response
We would like to thank the Reviewer for the valuable remarks and comments, which significantly helped us to improve our manuscript. Below we present point-by-point responses to the comments.
The overall presentation of the manuscript is correct, well written, and the references are updated. Since the authors only determined experimentally the log kw of the compounds, rather than discussing predictive ADME properties in so much detail, it is essential to include and discuss the experimental determination of other physicochemical properties that will substantiate with more confidence both the Lispinki rule and predictive ADME properties.
Other physicochemical properties have been previously determined and written in the following publications (ref. 25-27 in the manuscript). Relevant information has now been entered into the manuscript.
Minor issues
Abstract: line 12 – please change to “biological activities”
It has been corrected in the manuscript
Authors should put the bacteria and fungi species names in italics.
It has been corrected in the manuscript
Introduction lines 79-80: Authors said: “...because poor ADME properties are the main reason for failures and high costs in the design of new drugs [23]”. For many years this statement was true. However, currently, it no longer corresponds to reality. Currently, the main causes for failures are lack of efficacy and toxicity. Authors should update this statement and include a more recent reference.
It has been corrected in the manuscript
Results and Discussion: line 130. It is not clear what the authors mean by “…compound with the least developed structure.” To avoid misunderstandings, this sentence should be changed.
It has been corrected in the manuscript
Materials and Methods: Authors should add, in both chromatographic methods, the number of tests that were performed for each of the mobile phases.
It has been corrected in the manuscript
Reviewer 2 Report
The presented manuscript is certainly of interest to readers of the special issue "Recent Advances: Heterocycles in Drugs and Drug Discovery 2.0" The authors described in detail the lipophilicity and some biological properties according to the calculations of 28 compounds of related structure. Experiments were carried out confirming the results of lipophilicity calculations.
When reading the manuscript, there were questions that I would like to clarify before accepting the publication.
1) The authors carried out a computational experiment on the absorption of synthesized substances by Caco-2 cells. What is the reason for choosing this cell line?
2) Figures 2, 5, 6, 7 need to be improved. Now the drawings are difficult to understand. It may be worth breaking it into figures a, b, c, etc. and add captions
3) Figures 3 and 4 need to be updated. Now the resolution is not high enough
4) It is highly desirable to conduct several experiments that would confirm the data of calculations in biological media
5) Is the mechanism of inhibition of CYP3A4 isoenzyme known for the considered substances?
Author Response
We would like to thank the Reviewer for the valuable remarks and comments, which significantly helped us to improve our manuscript. Below we present point-by-point responses to the comments.
The presented manuscript is certainly of interest to readers of the special issue "Recent Advances: Heterocycles in Drugs and Drug Discovery 2.0" The authors described in detail the lipophilicity and some biological properties according to the calculations of 28 compounds of related structure. Experiments were carried out confirming the results of lipophilicity calculations.
When reading the manuscript, there were questions that I would like to clarify before accepting the publication.
1) The authors carried out a computational experiment on the absorption of synthesized substances by Caco-2 cells. What is the reason for choosing this cell line?
We appreciate the Reviewer's comment. In this analysis, we chose Caco-2 cells because they are widely used in in vitro studies to predict and evaluate the oral absorption of new drugs due to their functional and morphological similarity to human enterocytes.
2) Figures 2, 5, 6, 7 need to be improved. Now the drawings are difficult to understand. It may be worth breaking it into figures a, b, c, etc. and add captions
3) Figures 3 and 4 need to be updated. Now the resolution is not high enough
We thank the Reviewer for this good point. All graph have been improved both in terms of their readability and graphic quality. We hope the revised version is clearer.
4) It is highly desirable to conduct several experiments that would confirm the data of calculations in biological media
We thank the Reviewer for this important point. We agree that experimental studies are important and do not always support computational data. Therefore, we plan to conduct such research. Due to their time-consuming nature, this will be the subject of another publication.
5) Is the mechanism of inhibition of CYP3A4 isoenzyme known for the considered substances?
The mechanism of inhibition of the CYP3A4 isoenzyme is not known for the tested compounds, because the results obtained for the presented compounds were performed using software in which the ability of the compounds to inhibit cytochrome P450 enzymes was used in the computational model, without taking into account the mechanism.
The available literature also does not describe the mechanism of inhibition of the CYP3A4 isoenzyme by structurally related substances.
Round 2
Reviewer 1 Report
Authors have modified the original submission, addressing the issues raised during the last revision.
Reviewer 2 Report
The authors addressed all comments and I recommend now accept in present form